# PRETRAIN KNOWLEDGE-AWARE LANGUAGE MODELS

## ABSTRACT

How much knowledge do pretrained language models hold? Recent research observed that pretrained transformers are adept at modeling semantics but it is unclear to what degree they grasp human knowledge, or how to ensure they do so. In this paper we incorporate knowledge-awareness in language model pretraining without changing the transformer architecture, inserting explicit knowledge layers, or adding external storage of semantic information. Rather, we simply signal the existence of entities to the input of the transformer in pretraining, with an entity-extended tokenizer; and at the output, with an additional entity prediction task. Our experiments show that solely by adding these entity signals in pretraining, significantly more knowledge is packed into the transformer parameters: we observe improved language modeling accuracy, factual correctness in LAMA knowledge probing tasks, and semantics in the hidden representations through edge probing. We also show that our knowledge-aware language model (KALM) can serve as a drop-in replacement for GPT-2 models, significantly improving downstream tasks like zero-shot question-answering with no task-related training.

## 1 INTRODUCTION

The strong effectiveness and rich generalization ability of pretrained language models (PLMs) (1; 2; 3; 4; 5) have raised many questions about what is captured in transformer networks and why. Recent explorations found the pretrained language models may "rediscover" the linguistic pipeline at various transformer layers (6), can serve as implicit knowledge bases for relation extraction (7), perform soft reasoning tasks (8; 9), and conduct some language tasks reasonably in a fully unsupervised, zero-shot, fashion (3; 10). With sufficiently large amount of parameters, i.e. several billions, and enough task-specific supervision, the pretrained language models can even directly generate answers for natural language questions, at the same accuracy with state-of-the-art reading comprehension systems, without using any context documents or knowledge graphs (11).

Impressive as they are, language models are still far from ready to serve as an "unsupervised multi-task learner" that learns knowledge directly from human language and generalizes to downstream language tasks (3). There are notable gaps of language models' performances on downstream tasks between models with (11; 12) and without (3) large amounts of task-specific fine-tuning. The language models still (de-)generate dull, factually-incorrect, or dream-like text when used in natural language generation (13; 14; 15; 16). These challenges often necessitate over-parameterization (17), grounding on external structural semantics (14; 16; 18; 19), or large amount of task-specific fine-tuning (11), which are costly, complicated, and not always feasible for every language task.

One potential limitation of these language models is their style of pretraining, e.g, auto-regressive language modeling (3) or masked language modeling (1), wherein transformer networks process a sequence of words and are asked to predict the next/masked words. There is no explicit guidance to the transformers that humans prefer them to capture correct, real-world information. As a result, all the knowledge captured in these pretrained language models is only signaled by patterns of co-occuring words in the input sequence that is learned implicitly during pretraining.

In this paper, instead of creating bigger models or adding knowledge-specific architectures, we propose to more efficiently leverage the existing parameters in the standard transformer language model, by simply making them aware of the various forms an entity can manifest itself as, and its role in the surrounding text. More specifically, this knowledge-awareness is communicated via the input fed to PLMs and in the output expected from them during pretraining. For input-awareness,

we use an entity-name (surface form) dictionary that tokenizes word spans to their most popularly referred-to entity, e.g., as fuzzy frequency-based entity annotations (20), and serve these entity tokens as a parallel input channel along with the word tokens. For output-awareness, in addition to the language modeling objective, we add an entity prediction task that guides the model to distinguish the correct entity from various negative distractions. The two objectives together explicitly guide the language model to predict not only the correct words, but also the correct entity behind those words during pretraining, without changing the network architecture.

By adding knowledge awareness to GPT-2 style auto-regressive language models, our pretrained language model, "Knowledge-Aware Language Model" (KALM), shows significantly improved handling of knowledge-sensitive tasks. In the LAMA knowledge probing tasks (7), KALM outperforms its entity-unaware baseline, GPT-2, by about 25% across all tasks at both base and large transformer sizes. Our 24 layer KALM (Large) is even comparable with the 17 Billion parameter GPT-2 on some tasks. It more accurately captures commonsense knowledge, factual semantics, and also relation semantics in these LAMA tests. The knowledge signals also aid generic language understanding: we have observed better language modeling perplexity and word prediction accuracy with KALM too.

The advantages in language modeling also transfer to downstream tasks. In zero-shot question answering, the exact match accuracy of the answers generated by KALM are 20%-100% better than those of an equivalent GPT-2 model. We did not use any task-specific supervision or additional gradient updates, relying solely on the unsupervised knowledge learned in KALM. We only feed in a few example question-answer pairs as templates to format how generated answers should look. Injecting rich knowledge signals leads to improvements approximately equal to those gained by doubling the transformer layers, indicating that PLMs can be trained more efficiently – growing the parameters exponentially is not the only way to improve language understanding.

To better understand pretraining and the advantage of knowledge awareness, we leverage the edge probe technique (6; 21) and dissect what is learned in the representations at various gradient step numbers throughout pretraining. We observe that the auto-regressive transformers start to learn the basis of language in the beginning of the pretraining, and gradually learns more complex semantics in the process; adding knowledge-awareness greatly accelerates learning of higher level semantics, e.g., coreferences and entity types, and helps the model perform better in those more complicated tasks.

## 2    PRETRAINING KNOWLEDGE-AWARE LANGUAGE MODELS

In this section we first present preliminaries in language modeling and then how we add knowledge-awareness in their pretraining.

### 2.1    PRELIMINARY

In this paper, without loss of generality, we mainly focus on the auto-regressive language modeling. Considering the text $X$ as a sequence of tokens (words or sub-words): $X = \{w_1, ..., w_i, ..., w_n\}$, the classical unidirectional factorization of language probabilities (22; 23) describes:

$$p(X) = \prod_i p(w_i|w_{<i}),  \tag{1}$$

where $w_{<i}$ refers to all the tokens appear before $i$. This conditional probability can be parameterized in various ways. An effective choice is to use the uni-directional transformer, as done in GPT-2 (3): $p(w_i|w_{<i}) = \texttt{transformer}(w_i|w_{<i})$.

The language modeling task provides a large amount of data to pretrain very deep transformer networks (5; 24; 25). Scaling up the transformer parameter sizes will lead to significant improvements in the language model capability: with wider and deeper transformer layers, it is observed that transformer language models start to output more complicated semantics beyond lexical and syntactic patterns (6; 8; 7). On the other hand, a roughly log-linear relationship between transformer size and output quality has been established, e.g. doubling the quality requires ten times more parameters and training data (3; 17; 10). Even in industry, the marginal gain of increasing parameters will eventually be outweighed by the cost to train and serve such models.

## 2.2 KNOWLEDGE-AWARE PRETRAINING

As shown in Eqn. 1, the pretraining is solely at the token level. All the semantics in PLMs are captured by the transformer indirectly; there is no *explicit* requirement in pretraining to better capture knowledge – yet we expect them capture knowledge beneath the raw word sequences implicitly, e.g., to generate factually correct statements.

In this work we mitigate this discrepancy by making transformer networks aware of knowledge in language model pretraining. Instead of stacking more layers or adding external knowledge storage, we present a knowledge-aware language modeling (KALM) framework that packs more information into the same amount of transformer parameters. The first step to introduce *knowledge awareness* is an *entity tokenizer* that forms an additional entity token sequence (26) to signal the existence of entities in the *input* and *output* of the pretraining process.

**Entity Tokenizer.** An entity tokenizer segments the text sequence into entity ids using a surface form dictionary, which maps word-ngrams to entities: $w_{i:i+k} \xrightarrow{\text{dict look up}} e_i$, where $e_i$ is the most popular entity referred by the word k-gram $w_{i:i+k}$, and $e_i = \texttt{null}$ if $w_i$ is not part of any known entity surface names.

This simple dictionary look-up can be conducted efficiently, similar to the (sub)word tokenizer. Simultaneously, the text is tokenized into two channels – a word-entity duet token sequence (26):

$$X_{\text{duet}} = \begin{cases} \{w_1, ..., w_i, ..., w_T\} & \text{Word Sequence;} \\ \{e_1, ..., e_i, ..., e_T\} & \text{Entity Sequence.} \end{cases} \tag{2}$$

The two sequences are aligned position by position. If multiple (sub)words together form an entity name, the corresponding entity id is duplicated in each position corresponding to these words. For example, the name "United States" at $w_{i:i+2}$ is mapped to entity "USA" in $e_i$ and $e_{i+1}$.

Instead of enlisting a more precise entity linker or supervisions from entity labels (16; 18), a fuzzy frequency-based dictionary look up places higher expectations on the model to use clues in the text to jointly build token and entity representations that better reflect how language conveys knowledge. Using a highly tuned entity linker would propagate its own biases into the transformer.

**Knowledge-Aware Input.** Just as there is an input embedding for every word token, we allow the model to learn an entity embedding for each entity:

$$\vec{e_i} = \text{Embedding}_e(e_i) \in \mathbb{R}^{d_e}, \tag{3}$$

$$\vec{w_i} = \text{Embedding}_w(w_i) \in \mathbb{R}^{d_w}. \tag{4}$$

The two embeddings are combined to form the knowledge aware input:

$$\vec{t_i} = \vec{w_i} + \texttt{Linear}_t(\vec{e_i}), \ \texttt{Linear}_t \in \mathbb{R}^{d_e \times d_w}. \tag{5}$$

All the embeddings are randomly initialized and learned in pretraining.

**Knowledge-Aware Output.** The knowledge-aware input is fed into standard transformer layers, the same as the word-only input. Then in pretraining, besides the next-word prediction task, we also employ a next-entity prediction task to further incorporate knowledge-awareness.

Specifically, we use one output head for the word probability, one for the entity, and share all transformer layers between words and entities. If there are $L$ transformer layers and $h_i^L$ is the output of the final layer's $i$-th token, the loss for position $i$ is computed as

$$l_e(e_i|t_{<i}) = \max(0, \text{s}(\vec{h_i}^L, \vec{e_i}) - \text{s}(\vec{h_i}^L, \vec{e_-}) + \lambda), \tag{6}$$

$$\text{s}(\vec{h_i}^L, \vec{e_j}) = \cos(\texttt{Linear}(\vec{h_i}^L), \vec{e_j}), \tag{7}$$

$$\vec{h_i}^L = \texttt{transformer}^L(t_{<i}). \tag{8}$$

The transformers in Eqn. 8 are stacked multiple times similar to GPT-2. The scoring function in Eqn. 7 projects the hidden state into the embedding space with a $\texttt{Linear}$ layer and takes the cosine

similarity with an arbitrary entity $e_j$. Assuming that position $i$ is linked by our entity linker to $e_i$, we carefully choose a corresponding negative entity $e_-$ to contrast with $e_i$ using margin loss with margin $\lambda$ in Eqn. 6.

**Pretraining.** The knowledge-aware input and output are incorporated in the standard multi-task set up for our KALM knowledge-aware language model pretraining:

$$l_{\text{KALM}}(X_{\text{duet}}) = \sum_i l_w(p(w_i|t_{<i})) + \alpha l_e(e_i|t_{<i}). \tag{9}$$

It combines the language modeling loss $l_w()$ – cross-entropy as in standard PLMs – with the entity prediction loss $l_e()$, where $\alpha$ is a hyper-parameter to balance the two.

**Inference.** At inference time – whenever generating output text – KALM use the word prediction head $p(w_i|t_{<i})$, which is consistent with GPT-2. The entity prediction task is an auxiliary task that guides the model to attend to the entity semantics during pretraining; in inference only the shared transformer representations is used upon the input word and entity tokens. Compared with the standard GPT-2 style PLMs, the architecture of KALM only differs with an enlarged tokenization vocabulary with additional entity tokens, and their entity embeddings before the input to the transformer network. The transformer architecture and its layer configurations are kept consistent.

## 3 PROBING LANGUAGE MODELS

This section describes the techniques we use to probe the knowledge capability incorporated in the weights of pretrained language models, including LAMA Knowledge Probing (7), Edge Probing (6; 21), and the zero-shot performance on downstream tasks (3).

**Knowledge Probe.** Petroni et al. (7) developed the LAMA knowledge probing test which evaluates whether the language model can predict the factually correct token in "fill-the-blank" cloze statements. For example, the model is considered to include the corresponding commonsense knowledge "isCapbleOf" if it can predict the token "fly" or "eat" for the cloze test "birds can ___".

The LAMA cloze statements were semi-manually constructed from four knowledge sources: Google-RE (Wikipedia Relations), T-REx (Wikidata Relations), ConceptNet (Commonsense Relations), and SQuAD (Questions on Wikipedia). The knowledge graph triples were transformed to statements using manually defined templates. The SQuAD questions are manually rewritten to statements. LAMA only keeps cloze statements where the missing word is a single token w.r.t the tokenizer of the model in question, to avoid the influence of decoding. Sometimes the missing word can appear in the middle of a sentence, but for auto-regressive language models such as GPT-2 and Transformer-XL (27), LAMA only evaluates on those at the end. We refer to their paper for more details (7).

**Edge Probe.** Besides looking at the token-level predictions, Tenney et al. (6; 21) develop the edge probing" tasks to study the information (e.g., syntactic, semantic, or long range structures) in the learned hidden representations. Similar to the evaluation protocol in representation learning (28), the edge probing technique uses the PLM's hidden representations on one or multiple text spans (e.g., $h_{i:i+k}$) as fixed feature representations to linear classifiers on various linguistic/semantic tasks – A better performance indicates stronger knowledge capability of the PLM in the corresponding task.

In total eight core NLP tasks are included in edge probing (21): part-of-speech tagging (POS), constituent labeling (Consts.), dependency parsing (Deps.), named entity typing (Entities), semantic role labeling (SRL), coreference (Coref.), semantic probe-role (SPR), and relation classification (Relations). The first four tasks are considered more local, syntactical, while the later four involve "higher-order" and/or long-range phenomena (6).

**Zero-Shot Evaluation.** Radford et al. (3) demonstrate that their GPT-2 language models can be viewed as "unsupervised multitask learners" that perform downstream tasks without using any task specific pretraining or fine-tuning (3). Their deep GPT-2 models, though far from fully-supervised PLM's performance, also provide meaningful zero-shot results on several downstream tasks, such as question answering, summarization, and machine translation. These zero-shot results show promise that one centralized deep model can conduct many real world tasks without requiring much human annotation or task-specific finetuning. The new GPT-3 is another step in this direction, but at the cost of 175 Billion parameters (10)). In the short-term, this zero-shot evaluation is a good strategy to directly evaluate the knowledge captured in language model parameters.

Table 1: Size of evaluation sets. The number of relation types in LAMA are in brackets. All evaluations are zero-shot on their official testing data.

| Dataset | Items |
|---|---|
| **Language Modeling** | |
| WikiText-103 (tokens) | 270k |
| Lambada | 5.1k |
| **LAMA** | |
| Google-Re | 4.6k |
| T-Rex 1-1 (2) | 937 |
| T-Rex N-1 (23) | 20k |
| T-Rex N-M (16) | 13k |
| ConceptNet (16) | 11k |
| SQuAD (Statements) | 305 |
| **Zero-Shot QA** | |
| Trivia QA | 11k |
| Natural Questions (Short) | 3.7k |
| WebQuestions | 2.0k |

Table 2: Specifications of language models: parameters in the network layers (**Net.P.#**), in embeddings (**E.P.#**), number of layers (**L.#**), and hidden **Dim**ension. GPT-2 from OpenAI is marked by (OAI). Note that embeddings are looked up in constant time; the network capacity are mostly defined by **Net.P.#** (17).

| Model | Net.P.# | E.P.# | L.# | Dim |
|---|---|---|---|---|
| **Base** | | | | |
| GPT-2 | 90M | 38M | 12 | 768 |
| KALM | 90M | 458M | 12 | 768 |
| **Large** | | | | |
| GPT-2 (OAI) | 304M | 51M | 24 | 1024 |
| GPT-2 | 304M | 51M | 24 | 1024 |
| KALM | 304M | 471M | 24 | 1024 |
| **eX$^n$tra Large** | | | | |
| GPT-2 XL (OAI) | 1.46B | 80M | 48 | 1600 |
| GPT-2 1.5B | 1.46B | 80M | 48 | 1600 |
| GPT-2 17B | 16.9B | 214M | 78 | 4256 |

We focus on the zero-shot QA setting in Radford et al. (3), and include all the three datasets used by Roberts et al. (11): Trivia QA (29), Natural Questions (30), and Web Questions (31). In zero-shot setting, the PLMs are directly asked to generate the answer for the question; no task related information is provided for training, nor any context documents or knowledge graph of these questions are used. *All the knowledge has to come from the model parameters optimized on the pretraining corpus alone.*

The only additional information available to the PLMs is the task format as context during inference. Several example question-answer pairs were concatenated to the front of the question, to notify the model to generate answers (3). In total we use eight manually written dummy QA pairs (listed in appendix), e.g. "Question: how many wheels does a semi truck have? Answer: 18", and prepend them to each question of the testing dataset. All our models take the input in the format of

```
Question:  dummy Q\n Answer:  dummy A\n\n Question:  testing Q\n Answer:
```

and generates an answer without any additional training besides knowledge-aware pretraining.

## 4 EXPERIMENTAL METHODOLOGIES

**Pretraining with Knowledge Awareness.** All models in this study are pretrained on a dataset similar to OpenWebText[1]; it contains about 30 billion tokens after filtering and deduplication.

The entity tokenizer in KALM uses the entity dictionary from the CMNS Linker (20), which was harvested from the FACC1 and FAKBA entity annotations (32). We keep surface forms with frequency at least 1k, and entities with at least 100 links – in total about 1.4M entities. The tokenizer is forward-greedy by mapping the longest (at most 4-grams) surface form to its most popular entity (20). We choose this frequency-based dictionary lookup to favor higher entity coverage, linking speed, and recognition of entity polymorphism, while relying on the language model pretraining to handle any inaccuracy in the linking (26).

**Evaluations.** The language modeling evaluations include perplexity on WikiText-103 (33) and accuracy in LAMBADA last-word prediction (34), implemented the same as Megatron-LM (24).

The LAMA knowledge probes use their official setting (7)[2]. Particularly we use their set-up to evaluate auto-regressive LMs such as GPT (7). Our results are directly comparable with GPT-2 but not BERT as the two use different tokenizers. In *zero-shot QA* evaluations, the official evaluation splits of Trivia QA (29), Natural Questions (Short Answer Setting) (30), and Web Questions (31) are

---

[1]https://github.com/NVIDIA/Megatron-LM/tree/master/tools/openwebtext
[2]https://github.com/facebookresearch/LAMA

Table 3: Results on language modeling tasks and LAMA knowledge probing tasks. The LAMA numbers are average Precision@1.

| | Language Modeling | | LAMA Knowledge Probing | | | | | | |
| | Wiki-103 | Lambda | G-Re | | T-REx | | | C-Net | Squad |
| Model | Perplex. | last word | Total | 1-1 | N-1 | N-M | Total | Total | Total |
|---|---|---|---|---|---|---|---|---|---|
| **Base (∼100M)** | | | | | | | | | |
| GPT-2 | **20.85** | 33.73 | **3.99** | 24.17 | 14.19 | 16.72 | 15.66 | 7.67 | 4.55 |
| KALM | 22.51 | **40.26** | 3.27 | **44.70** | **24.95** | **25.07** | **25.96** | **8.61** | **6.64** |
| **Large (∼300M)** | | | | | | | | | |
| GPT-2 (OAI) | 22.50 | 42.98 | 3.71 | 56.03 | 19.77 | 19.93 | 21.6 | **10.86** | 8.04 |
| GPT-2 | 20.46 | 42.63 | 4.90 | 45.95 | 19.28 | 18.59 | 20.31 | 9.72 | 5.94 |
| KALM | **17.05** | **49.14** | **5.41** | **63.18** | **25.74** | **27.15** | **28.12** | 10.70 | **11.89** |
| **eX$^n$tra Large (>1B)** | | | | | | | | | |
| GPT-2 1.5B (OAI) | 17.37 | 51.23 | 4.30 | 62.31 | 21.53 | 19.61 | 22.77 | 12.28 | 11.54 |
| GPT-2 1.5B | 14.68 | 56.72 | 6.48 | 65.04 | 24.04 | 21.51 | 25.06 | 12.79 | 11.54 |
| GPT-2 17B | 10.21 | 67.98 | 8.77 | 76.82 | 29.60 | 27.14 | 30.95 | 14.39 | 22.38 |

used. Besides exact match (EM), we also show whether the generated answers contains the ground truth answer (cover-EM). The *edge probe* tasks use their official implementation (21)[3]. We probe the last layer of PLMs at every 1k pretraining gradient steps from 5k to 10k, and again at convergence.

**Compared Methods.** We mainly compare with the autoregressive GPT-2 architecture, and pretrain our own version GPT-2 in-house, following the implementation of Megatron-LM (24). All else remaining equal, we pretrain KALM exactly as in-house GPT-2 networks, except with knowledge awareness. The statics of all the evaluation tasks are listed in Table 1. The architecture specification of these models are listed in Table 2. We measured that the KALM models had a forward-pass runtime that was a constant added to that of an equivalent GPT-2; doubling the number of layers does not change this constant look up cost.

**Implementation Details.** We trained our models on either 32 or 64 Nvidia v100 GPUs in DGX-2 pods for 3-6 days using the DeepSpeed ZeRO optimizer (35). In all cases, we used the same linear learning rate warm-up over 3.2k steps to a maximum of 1.5e-4 with batches of 512 sequences each of length 1024, with cosine learning rate decay to a minimum of 1e-5 over at most 300k steps. Models were trained until convergence around 200-250k steps (4-5 epochs). Dropout of 0.1 was applied on the input tokens and the attention layers; the model had $\ell$-2 regularization of 0.01. We initialized all matrices with a uniform normal.

For KALM we added an additional dropout on the linked entities, replacing them with the `null` entity 10% of the time. We also sampled negatives for the output entity prediction task according to the following regimen: 1% of the time the negative was the `null` entity, 49% it was a random entity chosen from among all 1.4M, and the remaining 50% was a "difficult" negative chosen from the 100 nearest Trans-E neighbors of the positive entity. All our models are pretrained from scratch with entity embedding dimension $d_e = 300$. More details can be found in Appendix.

## 5 EVALUATION RESULTS

This section present evaluations on language modeling tasks and the probe tasks.

### 5.1 LANGUAGE MODELING AND KNOWLEDGE PROBING RESULTS

In Table 3 we show both the WikiText (perplexity) and Lambada (last word accuracy), adjacent to the LAMA knowledge probe (precision at 1). In general, the in-house GPT-2 results, with more training data, are slightly better than their Open AI implementations of equivalent architecture. We also consistently see that KALM has around 15%-20% improvement on LAMBADA accuracy. More importantly, on LAMA which directly tests factual correctness, KALM consistently improves over GPT-2 by margins of 40-80%, even approaching performance of a GPT-2 model with 5-fold more transformer parameters.

---

[3]https://github.com/ jsalt18-sentence-repl/jiant

Table 4: Zero-shot question answering performance of different models for three different question answering benchmarks. EM is exact match percent, and cover-EM is the percent counting whether the correct answer is a substring of the generated answer. All generated answers were generated by a greedy decoding of at most 20 tokens.

| Model | Trivia QA | | Natural Questions | | Web Questions | |
|---|---|---|---|---|---|---|
| | EM | cover-EM | EM | cover-EM | EM | cover-EM |
| **Fully Supervised** | | | | | | |
| T5 Base (11) | 29.1 | n.a. | 27.0 | n.a. | 29.1 | n.a. |
| T5 Large (11) | 35.9 | n.a. | 29.8 | n.a. | 32.2 | n.a. |
| T5 11B (11) | 50.1 | n.a. | 34.5 | n.a. | 37.4 | n.a. |
| **Zero-Shot** | | | | | | |
| GPT-2 Base | 3.44 | 4.77 | 0.81 | 1.24 | 2.08 | 2.92 |
| KALM Base | **5.87** | **7.16** | **1.75** | **2.13** | **3.53** | **4.79** |
| GPT-2 Large | 7.32 | 9.05 | 3.48 | 4.26 | 4.79 | 6.20 |
| KALM Large | **11.68** | **13.34** | **4.34** | **5.07** | **6.56** | **9.48** |
| GPT-2 1.5B | 17.78 | 21.59 | 6.08 | 7.95 | 6.20 | 12.65 |
| GPT-2 17B | 42.32 | 47.56 | 14.34 | 17.30 | 12.15 | 21.68 |

The improvements from KALM are most noticeable on T-Rex, which tests the model's skill on hundreds of knowledge base relations like "X is owned by ___". The gain is even more significant on more complex relations (N-1 and N-M), where GPT-2 is confused the most. KALM Large even matches the accuracy of GPT-2 17B on the most difficult N-M relations, *using only 2% transformer parameters*. Simply by being aware of the existence of entities in pretraining, KALM enjoys significantly better effectiveness and parameter efficiency in learning how to capture factual knowledge in standard transformer layers.

## 5.2 ZERO-SHOT TASK PERFORMANCES

In Table 4 we compare the zero-shot question answering performance of GPT-2 and KALM, as well as T5 fine-tuned in a supervised fashion. Remarkably, even though the largest GPT-2 model was only trained on unsupervised text prediction tasks, it still performs nearly as well as the fully-supervised T5 11B model on Trivia QA, confirming that indeed, larger language models do capture some knowledge in an unsupervised way. Yet, the captured knowledge seems more on the "trivial side" as GPT-2 performs much worse on NQ and WQ, which are more similar to what real users ask in search engines. This observation still holds even in the newly-announced GPT-3 with 175 Billion parameters (10), which has 30 EM on NQ, versus 44.5 in the RAG model which uses "only" hundreds of millions parameters (12).

Though far from perfect, KALM significantly improves the zero-shot accuracy on all three QA datasets. It more efficiently packs practical knowledge into its parameters that can answer human questions: the 12L KALM Base is very close to the zero-shot accuracy of the 24L GPT-2 Large. This indicates that one day pretrained language models may be able to capture complex semantics in real world tasks without needing orders of magnitude more parameters.

KALM has greater capabilities without sacrificing speed. Its extra entity embeddings do not increase run time much – they only introduce constant time matrix lookup operations after a linear time entity tokenization – this is one of our motivations of first adding the knowledge "awareness" to standard transformers before proceeding to specialized knowledge layers or external "memory" modules.

## 5.3 PROBING KNOWLEDGE AWARE PRETRAINING

In Figure 1 we show the edge probing results during the language model pretraining in terms of F1 on the eight NLP tasks (21). Theses tasks are ordered following the observed "NLP Pipeline" order in BERT layers (6). Tasks thought to require more sophistication or long range semantics are shown further left. Intuitively, the "easier" tasks that focus on local syntax, e.g., POS, are learned earlier in the pretraining process, while the "harder" ones, especially those require long range dependencies, such as relation extraction and coreference, are captured later in the pretraining. Transformers seems to learn what words are before learning how they interact to form higher levels of meaning.

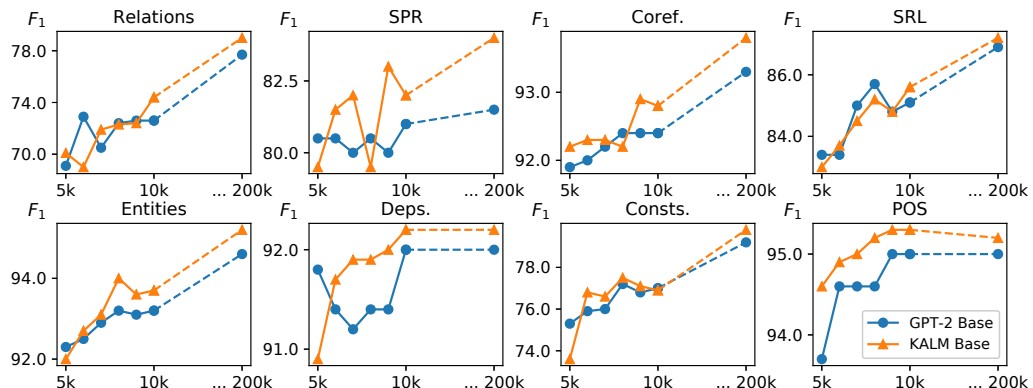

Figure 1: Edge probing results on eight NLP tasks. The base versions of GPT-2 and KALM are probed, along different training steps (x-axes), using micro-averaged F1 scores (y-axes).

The hidden representations learned by KALM are more informative in all eight categories at convergence. KALM starts slower in the beginning of pretraining (at 5k steps), as it needs to learn the additional entity embeddings from scratch; then the extra knowledge awareness quickly help KALM surpass GPT-2 after 5k steps. The benefits of KALM are more apparent on knowledge-related tasks, such as Entity typing, Relations, and Semantic Proto-Roles.

## 6    RELATED WORK

Are deep transformers just memorizing the corpus, e.g., using as many as one parameter per two training tokens (10), or do they abstract concepts and knowledge from the data (3; 5; 11; 36)? Many observed that the pretrained transformer weights do include certain forms of logic and semantic knowledge (6; 7; 9; 37): GPT-2 and GPT-3 can perform downstream tasks they never saw in training in a zero-shot fashion (3; 10); T5 with task-specific fine-tuning can accurately answer some questions without using the background document or knowledge graph (5; 11).

Yet it is also observed these enormous pretrained transformers often "dream" and produce language that might look right at first glance, but actually include biased (38), repetitive (13; 15), useless (39), or even worse, misleading information (14). To address this, previous research mainly grounds the language model to some forms of explicit semantic structures, for example, additional entity-oriented transformer layers (16; 18), or explicitly stored knowledge graph triples (19; 40; 41). Grounding provides a "safety net" for language models to look up a semantic facts in their external storage, instead of relying on their transformer parameters to incorporate correctness.

## 7    CONCLUSION AND FUTURE DIRECTIONS

KALM improves the knowledge learning capability of language models by signaling to the language model the existence of entities during pretraining. This simple knowledge-awareness significantly improves the parameter efficiency of transformers in pretraining. In our experiments on various tasks, KALM often matches the effectiveness accuracy of the "knowledge-unware" GPT-2 that uses twice as many transformer layers as KALM.

The fact that, adding knowledge awareness leads to improvements almost equal to those from doubling the transformer layers, indicates language models can be pretrained with better parameter efficiency. Brute-force blowing-up the model parameter numbers exponentially is not the only way to achieve a better language model – better intelligence is not only created by network sizes but can also by proper optimization, better understanding of pretraining, model architecture design, and the inductive biases within it.

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

# A APPENDIX

## A.1 MORE IMPLEMENTATION DETAILS

**CMNS Entity Tokenization** We performed entity linking in linear time with a forward-greedy approach. This means that the linker looks at all consecutive groups of up to four space-delimited words and repeatedly querying variants of that window in the dictionary (variants such as lowercase, uppercase, capital case, and without punctuation). We first query all the variants of the 4-word window before moving down to 3 words, and so on down to 1 word, before incrementing the window position and resetting it to 4. Whenever there is a hit in the dictionary, we link all the word tokens in the candidate mention span to the entity with the highest frequency, and then we move the window up past the words that were hit. Note that the entity linking is done in the "word space" meaning it looks at space-delimited words, but we take great care to ensure that all the tokens in the "sub-word" space that comprise the matched entity string are labeled as well. In all our experiments, we used a BPE subword vocabulary, but this technique will work for any standard word or subword vocabulary.

**Zero-shot QA Context Template.** We normalize both gold and predicted answers by lowercasing them. We filter Natural Questions to those with short answers as in the T5 paper (11): we ignore questions whose answers are longer than 5 tokens. We use the unfiltered version of the Trivia QA dataset. For both Natural Questions and Trivia QA datasets, we report the performance on the validation sets because the test sets with labels are not publicly available.

For each question in any QA dataset, we prefix the test question with a template of eight example questions. The exact template we use is:

```
question:  where is the Lincoln Memorial located?  \n answer:  Washington,
DC, USA \n \n '

question:  How long is the Nile river \n answer:  6250 miles \n \n

question:  who was elected president of the united states in 1928?  \n
answer:  Herbert Hoover \n \n

question:  what year did the September 11th attack occur?  \n answer:
2001 \n \n

question:  what elements of the periodic table are liquid at room
temperature?  \n answer:  bromine, mercury \n \n

question:  which pigment helps plant absorb energy from light?  \n answer:
chlorophyll \n \n

question:  who was the commander of the japanese navy for the majority of
World War II? \n answer:  Isoroku Yamamoto \n \n

question:  name of a famous highway without speed limits?  \n answer:
Autobahn \n \n

question:  how many wheels does a semi truck have?  \n answer:  18 \n \n
```

**KALM with Input entities only.** We found that a KALM model that only had knowledge awareness at the inputs (no entity contrasting loss at the output) had quality that was in between that of a knowledge-unaware GPT-2 and our full KALM. For a 12 layer input-only model, the wikitext perplexity was 26.45 and the lambada accuracy was 36.03. On the zero-shot question answering tasks, the triviaQA exact match was 4.5%, the natural questions EM was 1.5%, and lastly the webquestions EM was 2.7%.

