# OpenReview forum: "Pretrain Knowledge-Aware Language Models"
_ICLR.cc/2021/Conference — Reject_

### Official Review · AnonReviewer1 · 2020-10-28
**Official Blind Review #1**

**Rating:** 5
**Confidence:** 2

**Review:**

- Summary

This paper presents a knowledge-aware language model pretraining method without changing model architecture. Specifically, they add entity prediction task along with language modeling task to make the model aware of knowledge. Experiments show improved results on the LAMA knowledge probing task compared to GPT-2 models. They also show comparable results on zero-shot question answering task, even with only 2% transformer parameters compared to GPT-2 17B.

- Strengths

1. The authors propose a simple (i.e. without changing model architecture) method to give knowledge-aware signal during pretraining.

2. Proposed method consistently outperforms GPT-2 on LAMA and zero-shot QA.

3. Interesting to see that entities from fuzzy frequency-based matching can make LMs significantly better.

- Weaknesses & Questions

1. Need extra parameters to save entity embedding (~471M). However, as the authors mentioned, this can be viewed as an external memory module and embedding lookup doesn't need much computation.

2. I would like to see how much KALM can transfer to downstream tasks. The authors already show zero-shot QA results, but it would be better to see fine-tuning results on knowledge-intensive tasks. For instance, will KALM still outperform GPT-2 after finetuned on KILT tasks (https://arxiv.org/abs/2009.02252)?
(I know KILT released right before the due date of ICLR, but tasks in KILT are all already published)

3. Even though KALM outperforms GPT-2, it is still unclear for me the advantages of KALM compared to Entities as Experts (or EaE, https://arxiv.org/abs/2004.07202). For example, EaE outperforms KALM on the LAMA probing tasks, and EaE also outperforms T5-11B when finetuned.

4. The authors mentioned fuzzy frequency-based entity matching is better than precise entity linking (e.g. EaE) on page 3, but EaE outperforms KALM on LAMA probing tasks. Based on the experimental results, precise entity linking seems more effective to me. It would be helpful if the authors can provide a more thorough comparison with EaE.

- Typo?

Page 3, "United States" at $w_{i:i+2}$ --> "United Staes" at $w_{i:i+1}$ ?

=======================================================================

Post AR:

Thank you for preparing detailed responses, which helped me to clarify several questions. However, one of my major questions is still unclear.

Although 4 out of the 11 KILT tasks are already included in the main paper, most of them are LAMA knowledge probing tasks or zero-shot QA tasks. It is still unclear how much and how robustly KALM can transfer to other downstream tasks with fine-tuning (e.g. Wizard-of-Wikipedia, FEVER, QA with fine-tuning). For instance, CorefBERT paper (which uses a similar idea but as you mentioned it use bidirecitonal attention) shows its transferability on QUOREF, six extractive QA benchmarks, DocRED, FEVER, five coreference resolution benchmarks and GLUE, which can convince me to choose CorefBERT over BERT.

Experiments on the paper are promising but not diverse enough to make me choose KALM over GPT2. Thus, I would like to stick to my original score.

---

> ### Author Response · Authors · 2020-11-22
> **Response to AnonReviewer1**
>
> We would like to thank the reviewer for their detailed questions and comments. We will address the Weaknesses and Questions in the same order here:
>
> 1. This is true, the entity embeddings do amount to extra learnable parameters, but not all learnable parameters are treated “equally”, especially at runtime: parameters assigned to represent the vocabulary or entities are used “sparsely”, that is, only when they are identified by the tokenizer or linker. Adding more vocabulary items does not substantially increase runtime (only memory). However, the transformer layers (which act more as the “brain” to compute over the input embeddings) are used for every input sequence, and the more transformer layers, the longer the latency.
>
> 2. Yes we did not know about KiLT at the time we authored KALM, but it seems like there is substantial overlap between KiLT and (LAMA + our QA tasks). LAMA includes T-REx and RE (transformed into a language prediction task instead of slot filling). And we also did TQA, NQ and WebQuestions (the third not included in KiLT). Combined, our tasks cover 4 out of the 11 KiLT tasks, plus a few that aren’t in KiLT. Also it seems the KiLT paper does not address any GPT style models, only BERT and encoder-decoder style.
>
> 3. There are three primary differences between KALM and EaE we would like to highlight
>       - A) We wanted to study whether generative models like GPT-2 can benefit from entity information, since at the time studies had been done only on BERT-style and encoder-decoder style models, which have the advantage of bidirectional attention over the input. The difference between causal and bidirectional attention is not trivial: EaE with 367M params and even Bert Large do better than a 17B parameter GPT-2 model on LAMA and the QA tasks.
>       - B) We wanted to ascertain whether a custom-engineered “Entity Layer” built into the network is necessary, or rather if we can keep the transformer architecture unchanged but just signal the existence of entities in the input. We took the approach of Occam’s Razor and decided to investigate the latter for NLG/GPT style models.
>       - C) the EaE paper trains an entity linker and has a mention span loss. The influence of this component is not really explored in that paper, so we isolated it from our system altogether by using a fixed, deterministic lookup dictionary. A trained linker can probably disambiguate misspelling, etc more effectively than a dictionary lookup, but that is not the goal of our work.
>       - Taken together, points A-C here isolate that knowledge can be imparted to a generative language model without training a linker, without changing the transformer architecture or its runtime, and without the benefits of bidirectional attention. This has profound impact for large scale products and corporations: for instance, if one has custom Cuda kernels already built for a standard transformer to improve inference time, then we can use KALM as a drop-in replacement for GPT-2 in an open domain QA system to generate more natural sounding answers to questions. (These NLG systems and custom hardware/kernels do exist in commercial applications).
>       - Lastly, in the grand scheme of language models, we wanted to take the route of Occam's razor and show that there is “another way” to improve them without exorbitant scale-up or tailor-made intermediate layers that may not generalize to other scenarios or tasks. We found it satisfying to see that the transformer architecture is robust to handling additional knowledge signals without any modification. We hope this inspires more researchers to investigate how to convey to transformers more structural information in natural text, such as lists, paragraphs, co-references, embedded urls, etc that may not be aptly revealed in the cut-and-dry linear input sequence that transformers consume now.
>
> 4. Ah yes, “United States” is element w_i to w_{i+1} to match e_i and e_{i+1}. Thanks for catching that.

---

### Official Review · AnonReviewer3 · 2020-10-28
**entity information needs further explanation**

**Rating:** 6
**Confidence:** 2

**Review:**

The paper introduces to use additional knowledge to pretrain language models, which adds entity information to input and output. And the method improves the performance compared to the corresponding models without the knowledge. This is an interesting piece of work and worth publishing. However, I have some comments and questions about the article.

Since the main point of the paper is adding entity information to the models, the paper would be better to include more explanations about the entity information. For example, with the dictionary look-up, what is e_i exactly? Why did the authors use such knowledge rather than other knowledge like POS or so? Why do we need negative sampling? How many classes are there for entity?

The models enjoy more information provided by entity. Then what about more data to provide more information? How much can we improve the performance by adding entity information rather than additional data samples?

The authors said that the models were trained with entity information from scratch. Can we use pretrained models, since we can simply add entity information to the input and output layers? It seems like that the additional information can fine-tune the pretrained models.

What is the role of the dummy QAs? And how did the authors come up with the 8 dummy QAs?

---

> ### Author Response · Authors · 2020-11-22
> **Response to AnonReviewer3**
>
> We would like to thank the reviewer for their input. We will address their feedback in the same order it was mentioned:
>
> 1. We tried to keep the entity linking stage as simple as possible. For example, assume every word in this sentence is a single BPE token: “Joe Biden will be president of the United States”. w1 = Joe, w2 = Biden, etc to yield a word sequence of [w1, w2, …w9]. Say for instance that the surface form “Joe Biden" corresponds to entity 789 and “President of the United States” corresponds entity 4, then the entity sequence will be [e789, e789, null, null, e4, e4, e4, e4, e4]. In our reference dictionary, say surface form “Biden” is also mapped to the entity for “Joe Biden”. Hence “Biden will be president of the United States” will have entity sequence [e789, null, null, e4, e4, e4, e4, e4]. The reference dictionary was curated from search engine queries which clicked on specific entities. It is somewhat robust to misspellings and abbreviations, since those would naturally occur in user queries.
>
> 2. Regarding Part of Speech or other textual features, we figured that the position embeddings in GPT-2 would already model position information sufficiently. It was less obvious to us that GPT-2 could identify entities (groups of subwords) since the tokenization operation outputs subwords that have no relation to each other besides their ordering (i.e. there is no “grouping”).
>
> 3. We need negative sampling because we need to tell the model that this chunk of words “President of the United States” belongs to entity e4 as opposed to, say, entity e63 which could be entity for “Secretary of State”. If we had only positive samples, the model would be far less robust to noise in the linker or noise in the text.
>
> 4. We stated that there were ~1.4M entities in our reference dictionary. However, each entity could be expressed in many different ways such as “Joe Biden”, “Joseph Biden”, “Biden”, “Joseph R Biden Jr.”, etc. The purpose is to have the model learn that these different expressions all map to the same person, and since we have a lot of textual pretraining data, it is highly probable that all these surface forms would exist somewhere in the corpus.
>       - We propose that these signals help build a better language model: Say the model needs to complete the sentence: “Biden wore a mask in his acceptance speech in ____”. The model would need to know that 1) Biden is from Delaware, and 2) candidates make acceptance speeches in their home states. This information can be learned by GPT-2 if it sees enough pretraining text that could include “Biden is a Senator from Delaware” or “...Joe Biden’s home state of Delaware”, but we argue it can be learned more efficiently if we attach IDs to the entities “Biden” and “Delaware” and all other presidential candidates.
>
> 5. Related to 4 above, the question of “how much can we improve the performance by adding entity information rather than additional data samples” can be answered with an analogy: Say you wanted to train an image classifier on N images that are only in grayscale. Could you take the same N images and build a better classifier if they had full RGB color channels? Yes, the additional channels help the model generalize to the real world, similarly, we add an “entity channel” to the “vocabulary channel” for transformers in NLP. Our KALM and GPT-2 models were pretrained on the same N tokens of text (N = 30B) for the same number of steps, but KALM had the luxury of seeing an additional “dimension” of structure in those N tokens. Does that make sense?
>       - We agree that even more structure (like bullets, paragraphs, etc) in text will lead to better language models.
>
> 6. We tried taking a pretrained language model and then adding the randomly initialized entity embeddings into the new channel, but the model was very sensitive to such changes and diverged. Hence we decided that such information must be incorporated in the beginning.
>       - There are other works like KnowBERT (https://arxiv.org/pdf/1909.04164.pdf) which explore finetuning a model with entity information, but this work was 1) done on BERT and 2) we felt somewhat unnatural splicing a 24 layer into two parts, inserting a “knowledge transformer layer” at layer 13, and then stitching them back together. We felt that knowledge only needs to be signalled at the input to a model and at the output layer, and let the transformer learn for itself everything in between.
>
> 7. The role of the “dummy QA” pairs is simply to condition the model to get into the “mode” of question answering. We created them to mimic the format of the answers that seemed common across TriviaQA, NaturalQuestions, and WebQuestions. Sometimes answers were dates, locations, distances, numbers, etc. We wanted to the model to “get in the habit” of creating answers in those formats. The GPT-3 paper goes into much greater detail on the role of “dummy” or “few shot” inputs and how the quality improves with more examples.

---

### Official Review · AnonReviewer2 · 2020-10-28

**Rating:** 4
**Confidence:** 4

**Review:**

This paper proposes KALM - a knowledge-aware language model that incorporates entity information. Specifically, the method involves using a dictionary lookup to match n-grams to entities from a database. Then, each token is embedded into two vectors — one using the surface form, and one using the entity ID it is matched to (if there is a match). These embeddings are then added and passed into a Transformer model to predict the next word given the context. Experiments compare the model to GPT-2 and T5 (two state-of-the art Transformer-based LMs) and demonstrate the benefit of adding entity information.

Strengths: The idea of using entity information is certainly interesting and new. Most details of the paper are clear from the writing.

Weaknesses: The paper leaves open several questions that should really be explored and discussed. Given that the modeling contribution is simply adding an extra embedding layer, it would be good to have detailed analyses that provide more insights into the contribution itself. (see points 5,6,7 below)

Detailed comments/questions:
1. The details of the dictionary lookup are not entirely clear. Do you lowercase all tokens? Do you handle small misspellings in the tokens? Do you perform fuzzy matches?
2. Thanks for providing several implementation details. However, the values of hyperparameter $\alpha$ are not discussed. Does this require a lot of tweaking or is the method stable to this choice?
3. (minor) The ‘zero-shot’ experiments really seem to be ‘few-shot’ tests since you provide the model with 8 or so examples.
4. The metrics used for the LAMA knowledge probing should be explained in the paper once (at least expand the abbreviations so readers can understand).
5. Table 3: The parameter sizes indicated are misleading. Large KALM has close to 700M params and not 300M as indicated. Similarly base KALM has ~550M params not 100M.
6. Related to the above point, it seems like KALM base has more parameters than GPT-2 base (and even GPT-2 Large) and yet this aspect is not accounted for/investigated in the paper. In fact, the paper says “(KALM) more efficiently packs practical knowledge into its parameters … 12L KALM is very close … to 24L GPT-2 Large”. This seems to be very misleading information. Since Transformers are essentially a form of graph networks, I’m not sure entity embeddings can be treated separately from the model parameters since one can view those as edges part of the graph.
Concrete suggestion: Can you compare KALM vs GPT-2 (in all tables and Fig 1) while controlling for the #params? To me, the only closest comparison seems like KALM base vs GPT-2 Large where GPT-2 Large seems to be the better model?
7. I appreciate the authors efforts to test on a variety of benchmarks and I understand that training these huge models is not easy, but to me, the essence of the paper (i.e. addition of entity information) seems under-explored. For instance, could you perform ablations studies that only uses the entity embeddings in the input to see how much knowledge it provides? Another ablation can be on the matching done in the dictionary lookup (in terms of frequency cutoffs, n=1,2,3,4 in the n-gram matching, etc. ). One could also analyze the prediction accuracy of the model to see which types of entities (e.g. persons, organizations, etc.) work better vs those that don’t help as much.

-------
Update: Thanks to the authors for their response, which helped clarify several of my minor questions and I believe those can be revised with a writing pass. However, I still think this paper has two significant deficiencies:

1. The parameter size comparison still seems flawed to me. The authors say that one can discount the entity embeddings, but can we really? Aren’t they part of the model’s ‘representation’ even if the inference does not use all of them in a single forward pass? Several neural net architectures exist (including large-scale LMs) that do not use all inference paths but still count them in the total params. The BERT vs GPT-2 example provided in the rebuttal is only a difference of 26k tokens (still significant!) but here we're talking about a few hundred million parameters!
At the very least, I think the true sizes of the models must be acknowledged and one can add the point on entity embeddings vs 'brain' parameters as a caveat, but it seems scientifically inaccurate to me to claim otherwise. To be clear, I don't think the size issue detracts much from the main contribution of adding entity embeddings here (i.e. this work may still be of interest even if the size of model is larger), but the current version of the paper has several claims about size savings that seem incorrect.

2. The analysis of the proposed method (where it helps, where it fails, which hyperparams matter) is still lacking. The authors did mention one ablation in the supplementary that I missed, but I don’t think that is sufficient for a reader to understand how to build on this method in this future without re-running all the experiments, doing an extensive hyperparam search, etc.

---

> ### Author Response · Authors · 2020-11-22
> **Response to AnonReviewer2**
>
> Thank you for your review. We will address the reviewer’s specific questions in the same order:
>
> 1. The dictionary lookups follow a cascade of choices: first match all upper case (e.g. MSFT), then try to match camel-case, lower case. Misspellings are handled by the reference dictionary, which does contain some common misspellings that users issued to a search engine (e.g. Yoube -> Youtube). There is no fuzzy matching; the only other rule is we prefer to match longer surface forms over shorter ones e.g. "President of the US" as opposed to just "US". The purpose was to try to keep entity linking as simple as possible and decouple it from model quality: no learned entity linker, just the “wisdom of the crowd” from mined search engine queries.
>
> 2. Yes good catch, the value of alpha is stable as long as it is not too big relative to the language modeling loss - we used 1.0 but 0.1 also works. We designed our entity loss function to have a bounded range of essentially [-2, 2] +/- gamma thanks to the cosine scoring function (when the model is random, the loss is expected to be gamma, which was 0.25 in our case). Coupled with the well-understood behavior of language modeling loss (which, when the model is random is around 10-12 and decays nicely as seen in the GPT-3 paper), the combined loss is well behaved. Stability of loss functions is crucial in pretraining large models because there are lots of other sources of instability and training jobs can diverge frequently.
>
> 3. Yes, this work was written before the GPT-3 paper normalized a new definition of ‘few-shot’. To clarify, our experiments do use a few examples in the input context and are hence ‘few-shot’ w.r.t to GPT-3 definition. We did not apply any gradient updates from those examples.
>
> 4. A simple overview of LAMA: Google-Re and T-Rex are simple knowledge base triples like (Abraham Lincoln, born-in, Illinois) which were turned into text using templates, e.g. “Abraham Lincoln was born in Illinois”. Squad if formatted similarly. The prediction task for GPT-2 is basically to eliminate the object entity (Illinois) from the input and try to generate it from the vocabulary as whole. The metric is precision@1, basically, measuring whether the model’s highest scoring vocabulary item is the correct answer or not. There are a few details, such as all object entities must be a single vocabulary item (if Illinois -> “Ill - in - ois” with BPE encoding, then the whole triple is disqualified). This is one reason why it is difficult to compare BERT and GPT-2 on LAMA - they use different vocabularies. All the rows in table 3 compare exactly the same triples since each model uses the same vocabulary. LAMA results on BERT models will be over a slightly different set of triples due to its use of WordPiece instead of BPE.
>
> 5. (and 6) We tried to group parameters based on their function -- parameters assigned to represent the vocabulary (which act as “sensory inputs”) have an inherently different role than parameters in the transformer layers (which act more as the “brain” to compute over the sensory inputs). For example, BERT-large and GPT-2 are considered to be equivalent in size, even though BERT’s wordpiece vocabulary is 30k tokens, while GPT-2’s BPE is 56k tokens. However, they both have the same number of transformer parameters.
>       - While it is true that KALM Large has 700M trainable parameters, 471M of those are embeddings. The only parameters KALM gets to use at runtime/train time for a particular input sequence are the same as GPT-2: the 304M transformer network parameters (brain) plus the token embeddings plus our linked entity embeddings. It’s not like KALM leverages all the extra entity embeddings all the time; ~99% are ignored because they are not linked in a given input sequence. However, all the parameters in the "brain" _are_ used all the time for inference/training.
>       - The statement about transformers being graph networks is true, but the edge information is created only in the “brain” via the famous query-key-value attention operation-- that is where elements of the input are compared to one another. The vocabulary items themselves are just nodes, and a textual input sequence is just a list of applicable nodes. And we argue that we can just increase the number of nodes (or make a second “channel” of nodes) to make the model more sensitive to information about entities.
>
> 6. (see 5)
>
> 7. We agree, and we did explore the suggested first ablation study -- adding entity inputs but no additional entity training signal. We placed it in the appendix of the original paper,
>     > “KALM with Input entities only”
>
>     and found that the extra entity awareness did improve the model beyond that of a knowledge-unaware GPT-2, but it was not as good as KALM which had the advantage of more training signals. We can add this study as a row in Tables 3 and 4.

---

### Official Review · AnonReviewer4 · 2020-10-29
**Simple method to improve a language modeling knowledge about entities**

**Rating:** 7
**Confidence:** 3

**Review:**

(This review is a collaboration between a senior and a junior reviewer. The junior wrote most of the review, and the senior modified the review. Both reviewers read the paper in detail.)

Summary:

This paper presents an alternative pretraining technique for language models in order to make them more "knowledge aware". In addition to pre-training a model with the traditional language modeling objective, this work proposes an entity prediction task as pre-training. Furthermore, this work also incorporates entity tokens at the input level of the model by summing the word embedding and its corresponding entity embedding.

Experimental results show that such pretraining method yields models with greater factual knowledge according to the LAMA knowledge probing task compared to vanilla GPT2 models. In addition, it is also reported that such models perform better than a GPT2 model of the same size on TriviaQA, Natural Questions and Web Questions, in a zero-shot setting; and achieves competitive results when compared to larger models.

This paper is well written and easy to follow. This work is timely because it shows that increasing the model size and pre-training data size is not the only way to achieve strong performance on language related tasks (which is the current trend). Inductive biases like knowledge about entities can improve the performance of models to the point of achieving competitive results with bigger models.

Some limitations of this work include the lack of experiments on more diverse architectures like encoder-decoders to make sure that the proposed technique not only works on autoregressive GPT-type models. Moreover, only one approach for entity integration is explored.

Other than that, I have the following questions:

Interesting choice of the margin loss for entity prediction. More motivation should be added into why this loss instead of the cross-entropy loss. Optionally, an experiment comparing the performance of the model when trained with the cross entropy -vs- the margin loss would be interesting to have.

At inference time, you are forced to put entity labels on the input text as a preprocessing step. How much does this impact the generalization capacity of the model? How do you manage cases when an unseen entity is seen as input?

Are the experiments on QA really zero-shot? The examples presented in Appendix are not dummy as they reflect true questions and answers?

Minor:

Have you tried your pre-training technique on an encoder-decoder architecture like T5? It would be interesting to see that this technique not only works with auto-regressive models.

Have you tried to fine-tune your resulting pre-trained models on various tasks? It would be interesting to see if your pretraining method also helps finetuning more efficiently on some tasks. In addition, I would be curious to see if, while fine-tuning, the entity prediction accuracy and the LAMA probing performance of the model drops or remains more or less the same.

[1] Sun, Yu, et al. "Ernie: Enhanced representation through knowledge integration." arXiv preprint arXiv:1904.09223 (2019). [2] Ji, Yangfeng, et al. "Dynamic Entity Representations in Neural Language Models." Proceedings of the 2017 Conference on Empirical Methods in Natural Language Processing. 2017.


After author response:

Thanks for the response! After reading other reviews, I feel the novelty of this work is somewhat limited and it would be useful to highlight the differences with respect to previous work. Also a discussion on when your method is preferred over existing proposals. I am also decreasing my confidence score after reading other reviews.

---

> ### Author Response · Authors · 2020-11-22
> **Response to AnonReviewer4**
>
> We would like to thank the reviewers for their time and careful inspection of our paper.
>
> We do want to echo their analysis that
> > “Inductive biases like knowledge about entities can improve the performance of models to the point of achieving competitive results with bigger models.”
>
> To us, the “quality vs cost” curve is being pushed along a very expensive path by merely scaling up these models, but we can potentially get a “bigger bang for our buck” by signaling to the model information it may have otherwise ignored or didn’t know existed during pretraining.
>
> 1. Yes, while only one approach for integrating entities is explored, several other papers like EaE and KnowBert explore other variations, albeit for bidirectional attention models.
>
> 2. In our experience, the margin loss and cross entropy loss yield similar models, but cross entropy provides a more extreme “all or nothing” feedback signal. On the scale of 10^10 pretraining tokens and 10^6 entities in our entity prediction task, predicting the entity E_1 = “Hunter Biden” instead of E_2 = “Joe Biden” in the sentence “Biden seems to have won the US election” is far less egregious than predicting E_3 = “The Great Gatsby”. Cross entropy would treat both as equally incorrect, whereas margin allows for some flexibility, and in our experience, greater generalizability.
>
> 3. Great point, at runtime, the existing input text must be entity-linked to match what the model saw during training time. Because we dropped-out the entity links during training, the model is somewhat resilient to missing entity signals for short texts. The model does degenerate to actually being worse than GPT-2 when there are no entity signals at all for longer texts, since again, it was trained to expect a decent amount of those signals.
>
> 4. Yes, this work was written before the GPT-3 paper normalized a new definition of ‘few-shot’. To clarify, our experiments do use a few examples in the input context and are hence ‘few-shot’ w.r.t to GPT-3 definition. We did not apply any gradient updates from those examples. AFAIK, it used to be the case that “zero shot” learning meant doing a task without any gradient updates. No the dummy examples do serve a purpose, which is to condition the model to answer questions in a concise format such as a date, number, length, person, etc just like Natural Questions short answers are written.
>
> 5. We did not have the resources at the time to pretrain different architecture like encoder-decoder.
>
> 6. Future work will show finetuning, but we wanted to expose the model’s quality without any confounding factors such as additional training data, etc, and just use the zero-shot capabilities.

---

### Decision · Program_Chairs · 2021-01-07
**Final Decision**

**Decision:**

Reject

**Comment:**

The authors propose to improve the LMs ability on modelling entities by signalling the existence of entities and also allowing the model to represent entities also as units. The embeddings of the surface form and then entity unit are then added and passed through a layer to predict the next word.
The paper evaluates on QA and conducts probing tasks and shows that such an entity modelling results in better performance.

All reviewers have found the idea conceptually simple and novel. At the same time a number of concerns are raised, with the most important being the lack of understanding around which and how hyper-parameters matter for this model and, most importantly, the confounder introduced to the model by the much larger size of parameters introduced by the embedding layers. While the authors comment that not all the parameters are used all the time, the size of the embeddings still count at the total size of parameters a model has. Thus, without properly controlling for this (e.g., have an another model where the extra embedding params are given to another part of the model), it is difficult to determine whether adding more parameters was the solution or adding more parameters for modelling the entities.